# Investigating the Risk of Patient Manual Handling Using the Movement and Assistance of Hospital Patients Method among Hospital Nurses in Botswana

**DOI:** 10.3390/ijerph21040399

**Published:** 2024-03-26

**Authors:** Kagiso Kgakge, Paul Kiprono Chelule, Morris Kahere, Themba Geoffrey Ginindza

**Affiliations:** 1Discipline of Public Health Medicine, School of Nursing & Public Health, University of KwaZulu-Natal, Durban 4041, South Africa; paul.chelule@smu.ac.za (P.K.C.); ginindza@ukzn.ac.za (T.G.G.); 2Department of Health Promotion & Education, Boitekanelo College, Tlokweng, Old Naledi Kiosk, Gaborone P.O. Box 203156, Botswana; 3Department of Public Health, School of Healthcare Sciences, Sefako Makgatho Health Sciences University, Pretoria 0208, South Africa; 4Cancer & Infectious Diseases Epidemiology Research Unit (CIDERU), College of Health Sciences, University of KwaZulu-Natal, Durban 4001, South Africa; 21143420@dut4life.ac.za

**Keywords:** MAPO index, patient manual handling, low back pain, ergonomic risk, nurse

## Abstract

Background: Evidence on the prevalence of lower back pain (LBP) among nurses is widespread in the literature, with several risk factors being reported. These include manual handling of patients, repetitive bending and twisting movements, and long working hours. It is reported that LBP has negative health outcomes and causes poor work performance among healthcare workers (HCWs). The magnitude of ergonomic risks associated with these healthcare activities has not been adequately investigated in Botswana. Thus, this study aimed to investigate the ergonomic risk levels associated with the manual handling of patients and its association with the prevalence of LBP among nurses in Botswana. Methods: This was an observational cross-sectional hospital-based study conducted in a Botswana public tertiary hospital from March to April 2023. The Movement and Assistance of Hospital Patients (MAPO) tool was used to collect data on ergonomic risk levels. Data on the demographic characteristics of participants were collected using a tool adapted from the Nordic Musculoskeletal Questionnaire (NMQ). Odds ratios and 95% confidence intervals were estimated to determine the association between ergonomic risk levels and the prevalence of LBP. Results: A total of 256 nurses participated and completed the study. The self-reported prevalence of LBP in this study was 76.6%. The risk of acquiring LBP was high (90.5%) based on the MAPO index. Although the frequencies of self-reported LBP were high among nurses, these did not show any significant association with the MAPO index data. This could be partly due to the small sample size. Conclusions: There was a high prevalence of LBP in this study, which was corroborated by the MAPO index data. This has demonstrated the value of the MAPO index in forecasting the risk of patient manual handling. The findings might help Botswana formulate policies intended to address ergonomic preventive measures, directed towards reducing the MAPO index score by addressing the single risk determinants.

## 1. Background

Globally, LBP is increasingly becoming a major public health problem affecting individuals across all age groups from children to the elderly [1,2]. Numerous reports state that LBP is the leading cause of disability globally with approximately 619 million people affected by LBP at any given time. Furthermore, LBP is now the most reported musculoskeletal disorder (MSD), with a global lifetime prevalence of about 60–84% [2]. The prevalence of LBP is projected to continue rising, with Global Burden of Disease (GBD) 2020 suggesting that by the year 2050, 843 million cases of LBP will be reported [3], with major increases expected in Asia and Africa. LBP has also been reported to be associated with a high economic burden, which is majorly attributed to production loss due to work absenteeism and presenteeism, and healthcare utilization costs [2,3,4]. Despite the high prevalence and the reported burden of LBP, the mosaic of its pathophysiology is still far from being understood and remains largely unknown [2]. Evidence suggests that the etiology of LBP is multifactorial, with contributing factors from physical, chemical, and emotional stress [3,4,5]. Behavioral and environmental factors have also been correlated with the development of LBP among working-class adults [2]. Moreover, LBP has also been shown to strongly correlate with common mental health disorders. However, the causal relationship and the underlying mechanisms are still not fully understood [6,7]. Other risk factors such as increased body mass index [8], smoking [9], lack of regular exercise, increasing age, fear-avoidance beliefs, illness behavior, and work-related physical and psychological stress [7,10] have been reported.

The prevalence of LBP among nurses is reported to range from 33% to 90.1% [11,12,13,14,15] and peaks in middle-aged, working-class adults [5]. In comparison to other healthcare professionals like dentists, doctors, and pharmacists, the nursing profession has the highest prevalence of MSD [16,17]. It is reported that HCWs are susceptible to MSD due to their involvement in manual handling activities [16,18]. Varying prevalence of MSD among HCWs has been reported, with most studies showing that nurses are more susceptible due to their continuous exposure to environmental risks, including moving of patients. A study by Mbada et al. [16] revealed that nurses had the highest MSD prevalence of 88.4%, followed by doctors with 72%, physical therapists with 68.4%, and dentists with 50%. In Asia, the prevalence of LBP among nurses ranged from 32% to 59.8% [6,19,20], while in Europe, Switzerland, and India, LBP prevalence among nurses was 56.1% [21], 76%, and 32%, respectively [1,19]. A study conducted in Ethiopia also reported that nurses were the most affected among other HCWs, with an LBP prevalence of 52.8%, followed by physicians with a 19% prevalence [22]. The burden of LBP is comparatively higher in low-and-middle-income countries (LMICs), including Africa. In Sudan, an LBP prevalence of 87.5% was reported among nurses [15], which is comparatively higher than what has been reported in Asian, English, Indian, and European nurses. Other studies conducted among African nurses were carried out in Nigeria and reported the following prevalence: 77.1% [23], 73.5% [24] and 44.1% [18]. In Ethiopia, there was a prevalence of 76.1% [25], in Zimbabwe, 55.6% [26], South Africa, 58% [27], and Botswana, 68.6% [28].

Nurses are at a higher risk of developing LBP due to their involvement in the manual handling of patients [29,30] as a result of a lack of assistive devices such as hoist machines [25,31] and tasks that involve repetitive bending and twisting movements [14,25,26,31,32], putting additional strain on the lumbar anatomical structures [33]. Several studies on MSD and LBP among nurses have been widely investigated, but in Botswana, only one study was conducted by Kgakge et al. reporting a LBP prevalence of 68.6% [28]. Despite many studies reporting on the prevalence of MSD and LBP, relatively few studies have investigated the prevalence of LBP and its relationship with patient manual handling using the MAPO method. The few studies that have been published in this area using the MAPO method were conducted in European countries. Moreover, no studies have been conducted in Africa to establish the risk of patient manual handling, including in Botswana. To date, several techniques for evaluating and assessing patient manual handling have proven to be effective, with the MAPO method being one of them. MAPO was first used in 1999 [34] to investigate and assess the risk of patient manual handling. Moreover, MAPO is widely used in European countries such as France, Spain, and Italy. Thus, our study is probably the first in southern Africa to use the MAPO index method to evaluate MSD risks in the workplace and to reduce the burden of LBP among healthcare workers. Our hypothesis for this study was “The MAPO index is related to the incidence of low back pain, and raised index points correlate with the high incidence of LBP”. It is against this backdrop that this study was conducted to determine the current LBP prevalence and ergonomic risk factors associated with the manual handling of patients among nurses in a public tertiary hospital in Botswana.

## 2. Materials and Methods

### 2.1. Study Design

This is part of an earlier larger study that sought to determine the prevalence of LBP and assess and identify the workplace ergonomic risk factors associated with LBP. This was a descriptive, observational, cross-sectional health facility-based study. We conducted this study in compliance with the Declaration of Helsinki [35] and reported it in accordance with the Strengthening the Reporting of Observational Studies in Epidemiology (STROBE) guidelines [36].

### 2.2. Study Setting and Population

This was a single-center study conducted at PMH, the largest tertiary hospital located in the southern part of Botswana with a nursing population of just over 500. PMH is a referral hospital that offers a wide range of inpatient and outpatient services. This study was conducted from the beginning of March to the end of April 2023.

### 2.3. Participants and Eligibility Criteria

Participants were included in the study if they were professional nurses with at least 1 year of working experience at PMH and were willing to participate and sign the informed consent. Nurses with at least 1 year of experience were chosen due to their day-to-day involvement with manual handling tasks, and the reported high prevalence of LBP among this population category. Participants were excluded if their work experience was less than 1 year or they were unconsenting, pregnant, or had any other severe underlying medical condition. Wards were excluded if they did not admit any trauma patients from the accident and emergency wards, theatres, or outpatient departments.

### 2.4. Sampling and Sample Size Estimation

The initial stage involved purposive sampling and selection of the study site, and a census method was used to enroll study participants. The study site (PMH) was purposively selected because it is the largest hospital in Botswana with a wide variety of population demographics, which is somewhat representative of the study setting’s population. Mechanical hazards in hospitals include manual handling tasks (patients especially), which make the nursing profession one of the vulnerable occupations most affected by LBP. Since the sampling frame in this study involved approximately 500 nurses, a census approach was applied to recruit the study participants, as it was not practical to sample such a small population. Thus, all the eligible participants were approached to participate in the study if they consented.

### 2.5. Study Instruments

A self-reporting modified NMQ was used to collect data on participants’ demographic characteristics (age, sex, and years of service) and sought information on the prevalence, onset, severity, and duration of LBP and how it interferes with the activities of daily living (ADL). The NMQ is a validated and reliable tool for quantifying musculoskeletal pain in different body regions including LBP [37]. Workplace ergonomic risk factors were assessed using the MAPO tool [38], a standardized instrument that was tested for internal consistency and reliability in data collection. The MAPO tool is divided into two sections, where the first part collects information regarding the ward organization and training aspects of the nurses through an interview with the nurse in charge. The second part of the MAPO tool involves analysis/evaluation of the environmental hazards and equipment aspects through an onsite inspection.

The MAPO tool collects data on the following variables: the number of nursing staff involved in the transfer of patients, the type of patients and their level of disability, types of manual patient transfer operations performed, presence/absence of lifting devices, types of lifting devices used, and training of staff in the operation of these devices. This protocol permits the identification of all factors necessary for the calculation of the MAPO index, and these factors are the disabled patient/operator ratio, lifting factor, minor aid factor, wheelchair factor, environment factor, and training factor. A detailed description of these factors and how to calculate the MAPO index is found elsewhere [38]. The MAPO index is the exposure index that helps in the identification of LBP among nurses as a result of patient manual handling. MAPO index final scores are classified into three categories as follows: (i) 0–1.5 (mild, where the risk is almost negligible and the prevalence of LBP is almost similar to the general population); (ii) 1.51–5.00 (moderate risk, where medium- and long-term interventions should be implemented and the prevalence of LBP is 2.5 times higher than in the general population); and (iii) 5.01 and greater (high-risk exposure ward that requires immediate intervention plans, where the incidence of LBP is likely to be 5.6 times higher than in the general population).

### 2.6. Data Collection

A department-to-department onsite inspection and administration of the structured questionnaire was conducted. About 16 different wards with 256 exposed subjects were investigated for the prevalence of LBP and present environmental/occupational risks. The NMQ was administered by the principal investigator and research assistant to the nurses in all the included wards, while the data on exposure were collected by means of onsite inspection and interview with the nurses in charge and the technical staff in charge of safety in the hospital. Prior to the final data collection, all participants were provided with a detailed description of the aims and objectives of the study and the methods of data collection before informed consent was obtained.

### 2.7. Recruitment Strategy

The study was initially presented to PMH management and the heads of departments to gain entry permission into the wards. Participants were approached mostly during off-peak hours to avoid interruption of patient care.

### 2.8. Statistical Analysis

Data were processed and analyzed using the IBM Statistical Package for Social Sciences (SPSS) software version 27.0 for Windows (IBM Corp, Armonk, NY, USA). Descriptive statistics were used for participants’ demographic characteristics, with categorical variables expressed as frequencies and percentages and continuous variables expressed as means and standard deviations (SDs). The response variable, low back pain, was considered binary (presence of LBP or absence of LBP). The odds ratios and 95% confidence interval were estimated for the three increasing exposure levels based on the MAPO index score (mild 0–1.5, moderate 1.51–5.0, and high-risk exposure 5.1 and above). Crude odds ratios and adjusted odds ratios were calculated by means of univariable and multivariable logistic regression analysis, respectively. The adjusted *p*-value of <0.05 was deemed statistically significant. Data were checked and verified by the researcher and the statistician for completeness and to identify any possible errors and missing values prior to analysis.

### 2.9. Ethical Consideration

The study was approved by the Botswana Ministry of Health under the Health Research and Development Division (HPRD: 6/14/1) and the University of KwaZulu-Natal Biomedical Research Ethics Committee (BREC/00004365/2022). Gatekeeper permission was sought from the Princess Marina Ethics Committee and the management. Participation in the study was voluntary, and all participants were required to sign the informed consent prior to the final commencement of data collection.

## 3. Results

Table 1 presents the demographic characteristics of participants in the study which was conducted in March–April 2023. Out of the 348 total number of exposed participants, only 256 nurses successfully completed and returned the questionnaire, yielding a 73.6% response rate. When participants were asked if they experienced LBP in the past 12 months that lasted for 3 days or more, 76.6% of them were affirmative. The majority of the study participants were females (70.7%), with most of the participants (42.6%) being in the middle-aged category of 31–40 years. The study findings further revealed that the majority of the participants (32%) had less than 5 years of working experience as nurses.

The distribution of exposed participants in 16 different wards at the hospital is presented in Table 2. The results revealed that a high proportion of participants (19) were from the Male Surgical Ward (MSW), comprising 5 (25%) males and 15 (75%) females, followed by the Male Medical Ward (MMW) and Male Orthopedic Ward (MOW) with 19 participants each. The least number of participants was recorded in the Private Ward with only one (12.5%) male and seven (87.5%) females. The results on the prevalence of LBP among the 16 wards are also shown in Table 2, and when stratified by wards, the highest prevalence of LBP was recorded among males in the Female Medical Ward (43.8%) and Female Surgical Ward (38.9%); however, higher prevalence of LBP was observed among the females in the Private Ward (87.5%), followed by nurses working in the Pediatric Medical Ward (71.4%). Table 2 further explored MAPO exposure in the 16 wards, out of which 14 wards fell in the red band, with the highest scores recorded from the Post-Natal Ward, Male Orthopedic Ward, and Female Orthopedic Ward with scores of 27.5, 20.4, and 20.3, respectively. This indicates a high risk of developing LBP. In addition, two wards fell in the yellow band, Spinalis and ICU, with scores of 5.0 and 1.8, signifying a medium risk for developing LBP. None of the wards included in the study fell in the green band for negligible risk.

The risk factors associated with manual handling of patients in the 16 wards are tabulated in Table 3. As shown, lifting factors associated with manual handling such as hoists were not available in 93.7% of the wards; also, in all the wards (100%), minor aids such as sliding sheets, ergonomic belts, or sliding boards were absent or insufficient. The results of the study further revealed that in 75% of the wards, wheelchairs were absent, and only in 18.7% of the wards they were sufficient. Finally, the majority of the wards (75%), when assessed for environmental factors (which entailed structural features of the wards, toilets, and patient rooms), were better for participants to work with in terms of maneuvering nursing activities. Lastly, a higher proportion (93.7%) of the participants did not have training on specific ergonomic training operations, and only 6.3% received some information on certain operations concerning ergonomics operations.

To determine if the MAPO index score predicts the occurrence of LBP, we initially estimated the crude odds ratios by means of univariable logistic regression analysis, using LBP occurrence (0 for no LBP and 1 for the presence of LBP) as a dependent variable (Table 4). After the univariate regression analysis, all the following independent risk factors were not significantly associated with LBP (dependent variable): MAPO risk levels (OR = 1, 95% CI= 0.4–2.4, *p* = 0.9), gender (OR = 0.64, 95% CI = 0.34–1.18, *p* = 0.1), age, and work experience. We did not perform multivariate regression analysis because none of the variables’ *p*-values was *p* < 0.05. We performed a chi-square test of comparison to determine the bivariate association of the MAPO index risk level and the occurrence of LBP and estimated the associated effect size (Cramer’s V) to determine the strength of the association (Table 5). Based on the bivariate analysis, the MAPO index was not significantly associated with LBP occurrence (*x*^2^ = 0.012, df = 1, *p* = 0.978), and the effect size was weak (Cramer’s V = 0.02).

## 4. Discussion

This study marks the first attempt in Botswana to assess the prevalence of lower back pain (LBP) and the ergonomic risk factors associated with the manual handling of patients among nurses in public health institutions using the MAPO method. This novelty provides comparison opportunities with similar studies conducted internationally. The study reports an overall LBP prevalence of 76.6% within the past 12 months, aligning with similar findings in Ethiopia and Nigeria [24,25]. In contrast, studies in Hong Kong, France, Pakistan, Iran, and India showed lower LBP prevalence estimates among nurses, ranging from 32% to 54.9% [19,20,39,40]. Surprisingly, one systematic review conducted among different healthcare professionals showed that surgeons and dentists had the highest prevalence of LBP > 60% among their counterparts, which still remains lower than what this study found among nurses [40]. These variations may be linked to well-established healthcare systems and policies in those regions.

Some countries, such as Japan, Kenya, and Sudan, reported higher LBP prevalence rates above 80%, potentially attributed to inadequate healthcare resource allocation and a lack of prioritization of MSD health due to more urgent concerns with communicable diseases [15,41,42].

The study utilizes the MAPO method to analyze the risk of exposure to LBP among nurses. Surprisingly, all 256 nurses are identified as being at risk, with 90.5% at a higher risk. This deviates from similar studies reporting a fraction of nurses at high risk ranging from 20% to 60%, possibly due to poor infrastructural development and insufficient investment in public health protection [29,38,43,44]. The Post-Natal Ward stands out with the highest risk of exposure, boasting a MAPO index score of 27.5. This contradicts previous studies, where the highest recorded MAPO index value was 13.12 from pediatric wards, but aligns with other similar studies, which recorded similar MAPO index scores [29,30,45].

Remarkably, the Spinalis Ward, specializing in rehabilitative care for patients with spinal cord injuries, exhibits the lowest MAPO index of 1.8. The study attributes this to the meticulous considerations in room setups, including in-built shower chairs, rails, hoists, wheelchairs, transfer slides, and ample spacing in rooms and bathrooms. While some published studies [30,38,46] establish a significant association between the MAPO index and LBP, this study finds no statistically significant association. Nonetheless, our study validated the effectiveness of the MAPO method in predicting LBP, which was also found in other similar studies [46]. Bivariate regression analysis also fails to establish significant associations between age, length of work, and LBP, contrary to findings in other studies [18,30,47,48,49,50]. The inconsistencies in these results are attributed to the small sample size and the census method employed in the study.

Furthermore, the analysis showed that increasing age increases the likelihood of LBP occurrence with AOR: 4.8, 95% CI: 0.1–127.9. This prediction is corroborated by studies conducted by Awosan et al. and Guyton et al. who revealed that older age was associated with LBP [48,49]. Additionally, 26–30 years of experience was shown to be a better predictor of LBP occurrence level (AOR: 4.0, 95% CI: 0.3–91.5, *p* = 0.429). Our findings are consistent with other studies that demonstrated that the more years one works as a nurse, the greater the chance of contracting LBP [18], which could possibly be due to one being exposed to strenuous nursing activities over time such as long standing hours, heavy lifting of disabled patients and equipment, long shifts, and frequent awkward positions as they discharge their nursing duties. Surprisingly, other studies reported that younger nurses are more likely to develop LBP [18,50] than elder nurses due to a lack of knowledge in most nursing techniques and that the older nurses tend to shift more to administrative roles as they age and delegate most of the duties to the younger nurses.

Periodic risk assessments aim to systematically identify and document organizational risks in hospitals, facilitating predetermined occupational risk assessment for nurses. Hospitals need comprehensive risk inventories, and risk registers should document all identified risks for efficient management. Tailored intervention measures are crucial for addressing specific risks in various wards, emphasizing a better understanding of ergonomic factors to prevent MSD among nurses. Education and training in ergonomics are essential to mitigate nurses’ exposure to MSD risk factors, ensuring their occupational health. The Botswana Ministry of Health should develop ergonomic intervention programs, including multifaceted education on safe ergonomic principles like patient handling techniques and proper working postures. Periodic workshops conducted by trained ergonomic specialists can raise awareness of ergonomics and promote nurses’ well-being.

The use of technological lifting devices and minor aids is critical in reducing the risk of injuries during patient handling. Hospitals must be equipped with lifting devices, and manufacturers should provide intensive training on their usage. Sufficient wheelchairs are essential for patient movement, directly impacting the efficiency of quality patient care. Implementation of lift teams can assist in the manual lifting of heavy patients, complementing safe patient handling programs. Well-trained lift team members can provide ongoing coaching and training and promote teamwork within and between departments. Organizational modifications, including addressing staff shortages through training and recruitment, are crucial to reduce nurses’ workload and prevent burnout. Retaining skilled nurses requires improving working conditions, greater remuneration, and providing opportunities for further studies. A zero-lifting policy is a comprehensive approach to prevent musculoskeletal injuries among nurses by eliminating manual lifting.

### 4.1. Implications of the Study

This study serves as a wakeup call for funders, policymakers, and other involved stakeholders who need to ensure that employees are safe in the workplace by making informed decisions that address risk determinants for LBP, such as introducing the zero-lifting policy, increasing the availability of lifting devices and minor aids, forming lifting teams that can assist in the lifting of heavy patients, and providing more intensive education and training for nurses on ergonomics at large.

### 4.2. Strengths and Limitations

To our knowledge, this is the first study in Botswana to assess the ergonomic risk factors associated with the manual handling of patients among nurses; therefore, it can provide the basis for making decisions to help reduce LBP by developing robust ergonomic programs. This was a cross-sectional study; therefore, it cannot be used to establish causal relationships. Also, it was a self-reported survey, which might have introduced some level of recall bias. The other limitation is that the study was conducted in one setting in Botswana, and the population was small; also, we did not sample since we used a census method, which then makes it difficult to generalize the findings. Hence, a large-scale study is recommended, using experimental methods to establish the root cause of LBP. Lastly, the MAPO tool does not take into consideration other factors such as psychological and organizational risk factors, which also play a major role in LBP.

## 5. Conclusions

The study revealed a high prevalence of self-reported LBP among nurses, which was corroborated by the MAPO index data. The study also demonstrated the usefulness of the MAPO index in the prediction of LBP, as all nurses in the current study were at risk of acquiring LBP. Therefore, our study has provided essential information that can be used in the development and planning of policies that target a reduction in patient manual handling by nurses such as ergonomic training, education, and the provision of automated lifting devices, wheelchairs, and minor aid equipment, which in turn will aid the reduction in LBP among nurses and other healthcare professionals.

## Figures and Tables

**Table 1 ijerph-21-00399-t001:** Participant demographic characteristics (N = 256).

Characteristic	Number (%)
**Sex**	
Male	75 (29.3%)
Female	181 (70.7%)
**Age category (years)**	
20–30	94 (36.7%)
31–40	109 (42.6%)
41–50	33 (12.9%)
51–60	17 (6.6%)
60+	3 (1.2%)
**Work experience (years)**	
1–5	82 (32.0%)
6–10	50 (19.5%)
11–15	55 (21.5%)
16–20	28 (10.9%)
21–25	17 (6.6%)
26–30	12 (4.7%)
31+	12 (4.7%)

**Table 2 ijerph-21-00399-t002:** Prevalence of low back pain in different wards and gender distribution.

Ward	Number of Nurses		Prevalence of LBP		MAPO Index
	Male, n (%)	Female, n (%)	Male, n (%)	Female, n (%)	
ICU	6 (33.3%)	12 (66.7%)	4 (22.2%)	9 (50%)	5.0
NNU	4 (22.2%)	14 (77.8%)	2 (11.1%)	10 (55.6%)	6.7
FMW	10 (62.5%)	6 (37.5%)	7 (43.8%)	4 (25%)	14.6
MMW	8 (42.1%)	11 (57.9%)	3 (15.8%)	8 (42.1%)	16.7
Gynecology	2 (12.5%)	14 (87.5%)	2 (12.5%)	11 (68.8%)	18.7
Spinalis	5 (55.6%)	4 (44.4%)	3 (33.3%)	4 (44.4%)	1.8
Oncology	6 (40%)	9 (60%)	5 (33.3%)	8 (53.3%)	14.5
Private	1 (12.5%)	7 (87.5%)	1 (12.5%)	7 (87.5%)	5.3
PMW	0 (0%)	14 (100%)	0 (0%)	10 (71.4%)	11.0
PSW	5 (33.3%)	10 (66.7%)	5 (33.3%)	9 (60%)	14.0
MSW	5 (25%)	15 (75%)	5 (25%)	13 (65%)	13.4
FSW	9 (50%)	9 (50%)	7 (38.9%)	8 (44.4%)	14.0
Post-Natal	3 (16.7%)	15 (83.3%)	3 (16.7%)	12 (66.7%)	27.5
Antenatal	1 (5.9%)	16 (94.1%)	0 (0%)	10 (58.8%)	18.4
FOW	4 (25%)	12 (75%)	1 (6.3%)	11 (68.8%)	20.3
MOW	6 (31.6%)	13 (68.4%)	5 (26.3%)	9 (47.4%)	20.4

**Table 3 ijerph-21-00399-t003:** Analysis of single risk determinant of patient manual handling.

Factor	Sufficient	Insufficient	Absent	Total N of Wards
Lifting factor (LF)	-	6.3	93.7	16
Minor aids (AF)	-	-	100	16
Wheelchair (WF)	18.7	6.3	75	16
Environmental (EF)	25	75	-	16
Training (TF)	-	6.3	93.7	16

**Table 4 ijerph-21-00399-t004:** Univariate and multivariable logistic regression: association of the MAPO index and occurrence of LBP.

Variable	LBP Occurrence		COR	95% CI	*p*	AOR	95% CI	*p*
	−ve	+ve						
**MAPO**								
0–1.5	-	-						
1.51–5.0	7	20	1 (ref)					
5.01–10	53	176	1.0	0.4–2.4	0.9	1.0	0.3–2.6	0.985
**Sex**								
Male	22	53	1 (ref)					
Female	38	143	0.640	0.34–1.18	0.1	0.5	0.2–1.0	0.090
**Age (Y)**								
20–30	24	70	0.686	0.05–7.9	0.7	1.6	0.0–66.8	0.780
31–40	19	90	0.422	0.03–4.8	0.4	1.6	0.0–54.1	0.784
41–50	12	21	1.143	0.09–13.9	0.9	4.8	0.1–127.9	0.347
51–60	4	13	0.615	0.04–8.7	0.7	0.4	0.0–7.3	0.550
60+	1	2	1 (ref)					
**Work Experience**								
1–5	22	60	1 (ref)					
6–10	9	41	0.606	0.2–1.4	0.243	0.6	0.1–2.2	0.461
11–15	10	45	0.909	0.3–2.4	0.850	0.6	0.1–3.3	0.614
16–20	7	21	0.584	1.1–2.2	0.850	0.2	0.0–2.1	0.215
21–25	3	14	2.727	0.7–9.3	0.432	5.5	0.3–2.1	0.233
26–30	6	6	0.909	0.2–3.6	0.111	4.0	0.3–91.5	0.429
31+	3	9	0.367	-	0.893	0.2	0.1–133.5	-

**Table 5 ijerph-21-00399-t005:** Chi-square analysis for MAPO index and occurrence of LBP.

Variable	Chi-Square	Degree of Freedom	*p*	Effect Size
MAPO	0.01	1	0.978	0.02
Sex	2.05	1	0.152	0.09
Age (Y)	5.65	4	0.227	0.14
Work Experience	7.28	6	0.295	0.16

## Data Availability

Data from this study are the property of the Government of South Africa and University of KwaZulu-Natal, thus it cannot be made publicly available. All interested readers can access the data set from the Chairperson of the South African Health Research and Ethics Committee (BREC) from the following contacts: The Chairperson of the South Africa Health Research and Ethics Committee, Email: hrkm@kznhealth.co.za, Tel: +27-(033)-395-2805. The Chairperson BIOMEDICAL RESEARCH ETHICS ADMINISTRATION Research Office, Westville Campus, Govan Mbeki Building University of KwaZulu-Natal, P/Bag X54001, Durban, 4000 KwaZulu-Natal, South Africa Tel: +27-31-260-4769 Fax: +27-31-260-4609 Email: BREC@ukzn.ac.za.

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
