# Peer review of "Investigating the Risk of Patient Manual Handling Using the Movement and Assistance of Hospital Patients Method among Hospital Nurses in Botswana"

_ijerph, 2024, doi:10.3390/ijerph21040399_

Round 1

Reviewer 1 Report

Comments and Suggestions for Authors

Dear authors,

The study addresses an interesting subject, the objective is clearly defined, the methodology followed described with a coherent statistical analysis. However, I have my doubts about the originality and contribution to the international scientific community. The particularity is that the target population brings its own environmental specificities and working conditions. However, the number of international studies on the subject (prevalence low back pain) is very large, and the analysis tools used in observational cross-sectional hospital-based studies are classic. I encourage the authors to consider the following recommendations and questions to improve the article:

-          Generally speaking, to consider a publication, it seems to me that authors need to emphasize the originality of their work and perhaps highlight one or two points to set it out from the very large number of works on the prevalence of low back pain at international level. The composition of your sample is interesting, and you could perhaps go in this way.

-          Some references on MSD risks and their prevalence could be added. Some studies have proposed a meta-analysis of total prevalence and prevalence by body areas for healthcare professionals worldwide. These works could be mentioned at the beginning of the introduction. This would show that other healthcare professionals are very widely exposed to the MSD risk and would reinforce the fact that the low back is probably the most exposed area.

Why not cite the work of Cancarella et al in 2019, which showed the study confirmed the effectiveness of MAPO method to predict low back pain in 26 Italian hospitals in the Apulia Region. The approach followed by the authors is close to that proposed in the present article. It seems essential to me to compare the proposed work with that of Cancarella et al (introduction section) and to discuss it (discussion section).

Cantarella C., Stucchi J., Menoni O., Consonni D., MAPO Method to Assess the Risk of Patient Manual Handling in Hospital Wards: A Validation Study, Human Factors The Journal of the Human Factors and Ergonomics Society, 62(7), 2019.

Why was the original MAPO index reference not mentioned in the article?

Menoni, O., Ricci, M. G., Panciera, D., & Occhipinti, E. (1999). The assessment of exposure to and the activity of the manual lifting of patients in wards: Methods, procedures, the exposure index (MAPO) and classification criteria. Movimentazione e Assistenza Pazienti Ospedalizzati (Lifting and Assistance to Hospitalized Patients). La Medicina Del Lavoro, 90, 152–172.

-          There is no hypothesis at the end of the introduction, and the proposed objective is too vague. Isn't one of your objectives to show the relevance of the MAPO index on a sample composed of women and men, with characteristics, experience and working hours specific to Botswana nurses?

-          How can this study shed new light on the existing literature?

-          The method is clearly described, however in order to provide the reader with all the information a brief presentation of MAPO could be proposed in the appendix.

-          On the basis of your sample estimate (367 nurses), can you explain or comment on the fact that you only had a sample of 256 nurses who responded to the questionnaires? What conclusions can you draw about the generalisation of your results?

-          Furthermore, your sample is 71% female and 29% male. What impact does this have on the proposed results in terms of their generalizability? What can you conclude about the results of the study? I think it's important to address these points in the discussion.

-          It is imperative to add in the discussion section recommendations in relation to the objectives of the proposed study.

Reviewer 2 Report

Comments and Suggestions for Authors

I have some comments to the authors. I hope that my suggestions will be held for improving this work.

1. Please provide full name for each abbreviation when used the first time, either in the abstract or main text.

2. keywords have to come from the mesh database.

3. the title of the paper should be more informative.

4. The authors should prepare linear and multivariate regression analyses of the variables.

5. The tables and figure have to be self-explanatory.

6. Please consider changing some tables to figures.

7. The discussion section has to be expanded, including more clinical aspects and comparisons with other researchers' points of view.

8. In the introduction, the authors should better describe the background, the rationale of the study, and the novelty' pay attention to literature data on similar studies (elemental analysis) published earlier. 

9. The paper was not prepared according to the journal's recommendation.

10. Sections 5 and 6 should be put into section 4.

11. Please divide the methods and results section into subsections with descriptive titles.

12. please consider citing doi: 10.12659/MSM.940213.

13. most of the references are older than the last 5-6 years; please change them.

14. Conclusion should correspond with the study aimed.

Reviewer 3 Report

Comments and Suggestions for Authors

Dear Editor and respected authors,

Thank you for the opportunity to review the manuscript entitled “The prevalence of lower back pain and its association with ergonomic risk levels associated with manual handling of patients among nurses in public tertiary hospitals in Botswana.”

Here are my comments and areas of improvement:

Lines 17-19: The magnitude of ergonomic risks associated with these healthcare activities have not been adequately investigated.

I think the topic is overwhelmingly investigated worldwide, but the authors should consider adding context to this statement. For example, they could add “in Botswana” to the end of this sentence.

Line 22: The MAPO is not clear. I suggest writing the full name of this scale.

Line 29: “these did not show …”, what does the word “these” refer to? Please clarify.

Line 29: “This could be partly …”, what does the word “this” refer to? Please clarify.

Since the abstract is a stand-alone piece, I suggest clarifying what does the MAPO index mean. It is not clear why you were so interested in this index.

Line 43: “with GBD 2020 suggesting …” what does GBD refer to?!

Please avoid using acronyms that have no clear meaning. You should explain any acronym at its first appearance in the text to avoid confusion.

Line 56: “has been reported in Asian, England, India, and European nurses.” This sentence is grammatical incorrect and requires a reference at the end.

Line 103: Sampling and sample size estimation, this section requires a revision. The sampling approach is not clear. The sample size calculation needs to be detailed. Which software was used? What are the other parameters?

Line 115: write the full name of the NMQ.

Line 178: “only 256 nurses successfully completed the questionnaire”

Based on your sample size calculations, you did not achieve the minimum required sample size 367 cited in line 113.  

Line 179-180: “When participants were asked if they had LBP, 76.6% reported that they had LBP”

This statement is vague. What was the exact question to draw this conclusion? Did you ask participants if they were experiencing LBP at the time of questionnaire completion? Over the last week? The last month? The last year?! Please clarify.

Line 180: “Majority of the study participants were females (70.7%)”

I am just curios since the majority of the studied sample were females, how did you control for conditions such as pregnancy, menstrual cycle, that are known to trigger LBP?

Line 230 Discission

Overall, the discussion is well-written although short. However, I do notice that several numbers have been cited in the discussion. Usually, the discussion section should be more of a “discussion” than a “results”. Make sure that you only cite numbers that are highly relevant to the discussion.

Another major limitation of the discussion is the lack of connection between LBP among nurses with other healthcare professions. For example, physiotherapists do similar tasks to nurses and I think comparing your results with the previous study included physiotherapists would add a context and value to your paper.

I suggest adding the following references to your discussion and write a small paragraph on the comparison between LBP reported in your study and other healthcare professionals:

1)      Khairy WA, Bekhet AH, Sayed B, Elmetwally SE, Elsayed AM, Jahan AM. Prevalence, Profile, and Response to Work-Related Musculoskeletal Disorders among Egyptian Physiotherapists. Open Access Maced J Med Sci. 2019 May 17;7(10):1692-1699.

2)      Mierzejewski M, Kumar S. Prevalence of low back pain among physical therapists in Edmonton, Canada. Disabil Rehabil. 1997 Aug;19(8):309-17.

All the best,

Comments on the Quality of English Language

The manuscript requires moderate English editing. 

Round 2

Reviewer 1 Report

Comments and Suggestions for Authors

Dear authors,

The article has been improved and many changes have been made, but for me, there are still two points to improve before publication:

1) The changes made in the introduction are not totally relevant (reference to a field outside nursing) and above all do not mention international reference works in the field (proofreading note 2). It might be relevant to use the work of Jacquier-bret et al 2023, based on a global review of MSD risks by body zone (nurses) applied to healthcare professionals. The following works could be cited as examples in different countries: Choobineh et al, 2006 , Munabi et al, 2014, Ribeiro et al, 2016, Yeung et al, 2005, Pugh et al, 2020, Kee and Sao, 2007 to demonstrate the relevance and originality of the work presented.

-          Jacquier-Bret J., Gorce P., Prevalence of Body Area Work-Related Musculoskeletal Disorders among Healthcare Professionals: A Systematic Review, International Journal of Environmental Research and Public Health, 2023, 20 (1), 841,

-          Choobineh, A.; Rajaeefard, A.; Neghab, M. Association Between Perceived Demands and Musculoskeletal Disorders Among Hospital Nurses of Shiraz University of Medical Sciences: A Questionnaire Survey. Int. J. Occup. Saf. Ergon. 2006, 12, 409–416.

-          Munabi, I.G.; Buwembo, W.; Kitara, D.L.; Ochieng, J.; Mwaka, E.S. Musculoskeletal disorder risk factors among nursing professionals in low resource settings: A cross-sectional study in Uganda. BMC Nurs. 2014, 13, 7,

-          Ribeiro, T.; Serranheira, F.; Loureiro, H.Work related musculoskeletal disorders in primary health care nurses. Appl. Nurs. Res. 2017, 33, 72–77.

-          Yeung, S.S.; Genaidy, A.; Deddens, J.; Sauter, S. The relationship between protective and risk characteristics of acting and experienced workload, and musculoskeletal disorder cases among nurses. J. Saf. Res. 2005, 36, 85–95.

-          Pugh, J.D.; Gelder, L.; Cormack, K.;Williams, A.M.; Twigg, D.E.; Giles, M.; Blazevich, A.J. Changes in exercise and musculoskeletal symptoms of novice nurses: A one-year follow-up study. Collegian 2021, 28, 206–213.

-          Kee, D.; Seo, S.R. Musculoskeletal disorders among nursing personnel in Korea. Int. J. Ind. Ergon. 2007, 37, 207–212.

2) The authors write on line 275-277: "Although low back pain is the most reported MSD among healthcare providers, particularly nurses, it seems that the prevalence of low back pain among physiotherapists remains relatively low, for example in a study conducted in Egypt, they recorded 68.8% (45)."  It might be relevant to use the work of Gorce et al 2023 based on a global study of MSD risks by body zone (lower back) applied to physiotherapists. This systematic review and meta-analysis would make it possible to propose other values for the prevalence of LBP and improve the discussion.

Reviewer 2 Report

Comments and Suggestions for Authors

The paper has been significantly improved.

Author Response

There were no comments from Reviewer 2